# The Influence of Pressure-Induced-Flow Processing on the Morphology, Thermal and Mechanical Properties of Polypropylene Blends

**Pengfei Li [1,†], Yanpei Fei [1,†], Shilun Ruan [2], Jianjiang Yang [3], Feng Chen [1,*] and Yangfu Jin [1,*]**

1   College of Material Science and Engineering, Zhejiang University of Technology, Hangzhou 310014, China; lpf940226@outlook.com (P.L.); fyp9394@outlook.com (Y.F.)
2   Department of Engineering Mechanics, State Key Laboratory of Structural Analysis for Industrial Equipment, Dalian University of Technology, Dalian 116023, China; ruansl@dlut.edu.cn
3   Shaoxing Polytech New Materials Technology Co., Ltd., Xinchang 312500, China; xclvtai@126.com
*   Correspondence: chenf@zjut.edu.cn (F.C.); jinyangfu@zjut.edu.cn (Y.J.)
†   These authors contributed equally to this work.

**Abstract:** The pressure-induced-flow (PIF) processing can effectively prepare high-performance polymer materials. This paper studies the influence of pressure-induced-flow processing on the morphology, thermodynamic and mechanical properties of polypropylene (PP)/polyamide 6 (PA6) blends, PP/polyolefin elastomer (POE) blends and PP/thermoplastic urethane (TPU) blends. The results show that pressure-induced-flow processing can significantly improve the thermodynamic and mechanical properties of the blends by regulating internal structure. Research shows that the pressure-induced-flow processing can increase the strength and the toughness of the blends, particularly in PP/TPU blends.

**Keywords:** pressure-induced-flow processing; polypropylene blends; morphology; mechanical property; thermal property

## 1. Introduction

Polypropylene (PP) is one of the most commonly used polymer materials due to its high heat distortion temperature, tensile strength and yield strength [1,2]. However, due to its low impact strength and poor weather resistance, the application of PP in the engineering field is limited. The existing methods to improve the mechanical properties of PP generally include branching [3,4], material blending [5–7], and polymer material blend [8–13]. Polymer blend materials are widely used because of their low cost, simple operation and high feasibility. In recent decades, the method of combining PP with another high-performance polymer material to improve certain properties of PP has been widely used. Researchers have widely used PP/polyolefin elastomer (POE) to improve the toughness of PP [14,15], and the research of TPU toughening is also under development [16,17]. At the same time, PP/polyamide 6 (PA6) blends have attracted many attention in the field of engineering plastics because of their easy processing, good mechanical and thermodynamic properties. There are many studies to add dispersed phase PA6 to improve the mechanical strength of PP [18–22]. However, the compatibility of PP and PA6 is poor, resulting in poor performance of blend materials. Although the traditional reactive compatibilization method can improve its microstructure and mechanical properties, the process is more complicated and expensive. To solve this problem, a relatively new polymer processing method called pressure-induced-flow (PIF) has attracted the attention of researchers [22–30]. PIF processing refers to the process in which the solid polymer matrix is forced to flow and deform in a restricted direction under high pressure and a certain temperature below the melting temperature. Similar studies on the pressure deformation of semicrystalline polymers have been reported successively [31,32]. Compared with other thermoforming, PIF is solid-state

compression molding, and its molding temperature is lower than the melting point of the material. Furthermore, different from other thermoforming, PIF has only one fixed flow direction when compressed into tablets, so it can produce a special layered structure and a large amount of crystal deformation, and through our research, we found that this structure and crystal changes can improve the performance of the material. Moreover, our group and other researchers have also proved that compared with other processing methods such as biaxial stretching, PIF processing can simultaneously strengthen and toughen polymers more effectively [33–38]. However, there are few studies on the processing of polypropylene by the PIF process, herein, this work focuses on the influence of the PIF process on the properties of polypropylene blends. Conventional methods are used to blend PP with PA6, POE or TPU separately to obtain three mixtures of PP/PA6, PP/POE and PP/TPU, and then use the PIF process to further process these blends. This paper studies the influence of PIF process on polymer microstructure, thermodynamic properties and mechanical properties.

## 2. Experimental

### 2.1. Materials

PP1250 produced by Formosa Plastics from Ningbo, China has a melt index of 25 g/10 min and a density of 0.9 g/cm$^3$, Honeywell's PA6 has a melt index of 35 g/10 min, American Dow POE8400, 30 g/10 min, TPU 5070 melt index is 6 g/10 min. Sigma-Aldrich's PP-grafted maleic anhydride (PP-g-MA, SA9100 mark) is used as a compatibilizer.

### 2.2. Blend Material Preparation

Before blending, PP raw materials were dried in oven at 80 °C for 6 h, PA6 raw materials were dried in an oven at 120 °C for 12 h, POE and TPU raw materials were also dried in oven at 60 °C for 6 h to eliminate the absorbed moisture. Through the Hack Torque Rheometer, PP was blended with PA6, POE, TPU in a ratio of 1:1, and 5 wt% of PP-g-MA was added as a compatibilizer to increase compatibility, respectively, and 5 wt‰ of antioxidant to prevent high temperature oxidation of PP during blending. Then the processing temperature of PP/PA6, PP/POE, PP/TPU is 230 °C, 180 °C and 170 °C respectively, and the rotor rotation speed is 60 r/min.

### 2.3. Pressure-Induced-Flow Process

We prepared a rectangular (50 × 30 × 6 mm$^3$) with a flat heating table and place it in a customized mold with a cavity of 150 × 30 × 2 mm$^3$, as shown in Figure 1. The we used a fully automatic tablet press, and PP/PA6 was press-formed at 150 °C and 11.5 MPa; PP/POE was press-formed at 140 °C and 11.5 MPa; PP/TPU was press-formed at 130 °C and 11.5 MPa. We kept the pressure for 5 min and then cool to room temperature. The samples that have undergone the pressure-induced-flow process are recorded as PIF samples, such as (PIF PP/PA6).

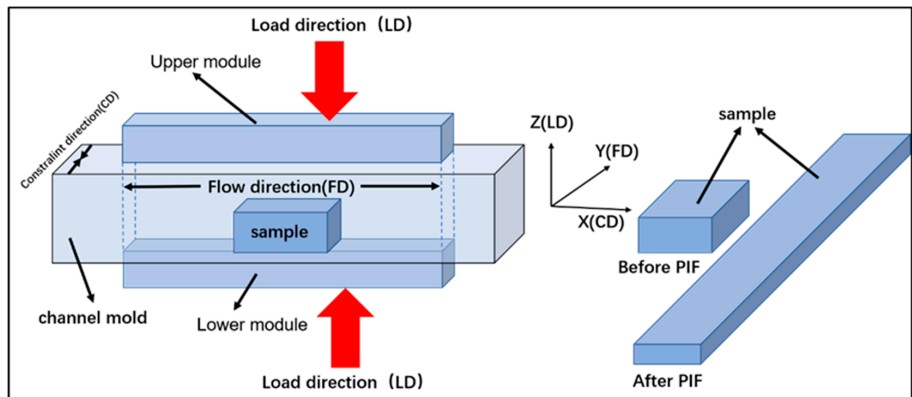

**Figure 1.** Pressure-induced-flow (PIF) process schematic.

### 2.4. Scanning Electron Microscope (SEM)

Scanning electron microscope (Philips XL30 from the Netherlands) was used to observe the fracture surface morphology of the samples before and after the PIF process. First, brittle fracture of the sample using liquid nitrogen, and then a layer of platinum was plated on the fractured surface of the sample to conduct electricity using an ion sputtering apparatus, and the scanning electron microscope observation was performed under an acceleration voltage of 10 KV.

### 2.5. X-Ray Diffraction (XRD)

An X-ray diffractometer (Bruker D8 Advance from Germany) was used to study the crystalline state of the samples before and after PIF processing under the condition of $2\theta = 5°\sim45°$ scanning rate of $10°/$min. The source of X-rays was Cu-Ka radiation with a wavelength of 1.54 Å. Scan under the working conditions of tube voltage of 40 KV and current of 50 mA. We scanned two vertical surfaces (XY plane and YZ plane as shown in Figure 1) then marked the XY plane as the 0° plane and the YZ plane as the 90° plane.

### 2.6. Differential Scanning Calorimetry (DSC)

Using DSC (TA Instruments Company, USA) in $N_2$ atmosphere, the thermal properties of the samples before and after PIF process were studied. The PP/PA6 sample was heated from 30 °C to 250 °C at a heating rate of 10 °C/min; the PP/POE and PP/TPU samples were heated from 30 °C to 200 °C at a heating rate of 10 °C/min. The melting temperature ($T_m$) and melting enthalpy ($\Delta Hm$) can be obtained through the scanning curve of DSC, so the crystallinity Xc (%) of the polymer can be calculated by formula (1) [39]:

$$Xc\ (\%) = \Delta H_m/(\Delta H_0 \times w) \times 100\% \tag{1}$$

where ($\Delta H_m$) is the specific melting enthalpy of the sample measured in the DSC experiment, ($\Delta H_0$) is the theoretical melting enthalpy of a 100% crystalline polymer matrix (209 J/g for PP, 190 J/g for PA6 [40]). Additionally, w is the mass fraction of PP or PA6 in the blend. When it is a pure polymer, w = 1.

### 2.7. Mechanical Test

The stress–strain curves for three-point bending test were obtained using Instron 5566 advanced materials testing system from the United States at room temperature and a loading speed of 5 mm/min according to the ASTM-D790 standard. The loading direction was parallel to the LD (Load direction). The notched Izod impact test was performed according to ASTM D256 using a TMI Izod Impact Tester. The impact direction was along the CD (constraint direction).

## 3. Results and Discussion

### 3.1. The Influence of PIF Process on the Microstructure of PP Blends

The low-temperature fracture morphology of PP/PA6, PIF PP/PA6, PP/POE, PIF PP/POE, PP/TPU, and PIF PP/TPU was observed by SEM as shown in Figure 2.

It can be seen from Figure 2A that the two-phase mixing state is better in PP/PA6 with the addition of a compatibilizer (PP-g-MA), and there is no phase separation. It can be observed from SEM that PA6 beads are evenly dispersed in the PP matrix, forming a sea-island structure. Figure 2B shows that the PP/PA6 blends have an orientation under the action of the PIF process, and the compatibility of the two phases is further improved. The layered orientation structure produced by PIF is helpful to improve the mechanical properties of the materials [33], at the same time, the further improvement of the compatibility of blends can also enable the materials to obtain better mechanical properties. In Figure 2C,E, it can be observed that the compatibility of the PP/POE and PP/TPU blends are very good, and there are almost no phase separation. Meanwhile, layered orientation structure can be clearly found from Figure 2D,F after PIF process.

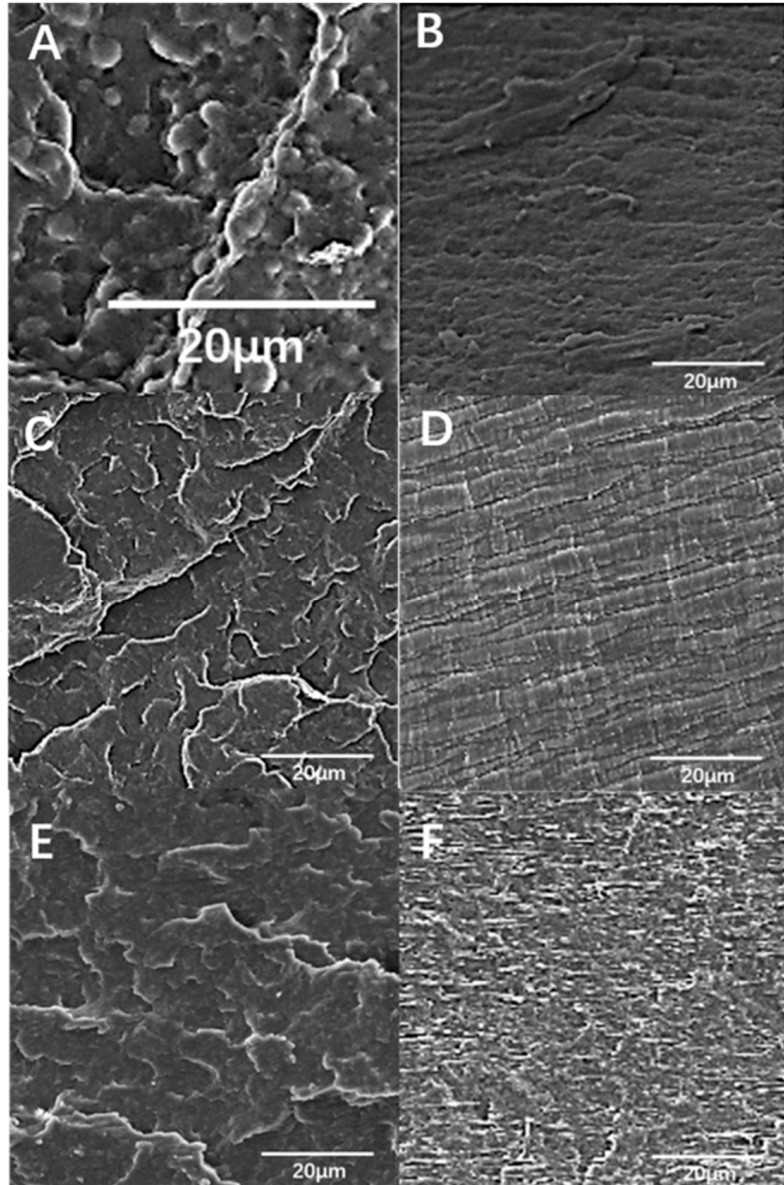

**Figure 2.** SEM morphology before and after the PIF process, (**A**) polypropylene (PP)/polyamide 6 (PA6), (**B**) PIF PP/PA6, (**C**) PP/polyolefin elastomer (POE), (**D**) PIF PP/POE, (**E**) PP/thermoplastic urethane (TPU), (**F**) PIF PP/ TPU.

Figure 3 is the XRD spectra before and after PIF process, we can observe that during the PIF process, the blends undergo uniform uniaxial deformation under pressure, the amorphous phase deforms along the flow direction, and the crystal phase also undergoes certain deformation under pressure, before PIF processing, it can be seen that the 0° (XY plane) scan curve and the 90° (YZ plane) scan curve almost overlap, as shown in PP/PA6 Figure 3A, PP/POE Figure 3C, PP /TPU Figure 3E. After PIF processing, it can be observed that the 90° (YZ plane) scan curve of PP/PA6 Figure 3B and PP/POE Figure 3D are low and the signals are weak. Even in PP/TPU Figure 3F, the 90° (YZ plane) scan curve is almost in a straight line with only faint peaks. These prove that the spherulites in PIF PP/PA6, PIF PP/POE, and PIF PP/TPU are oriented along the flow direction under high pressure, meanwhile, the amorphous phase is also oriented and deformed along the flow direction. The statistical results of XRD spectra are shown in Table 1.

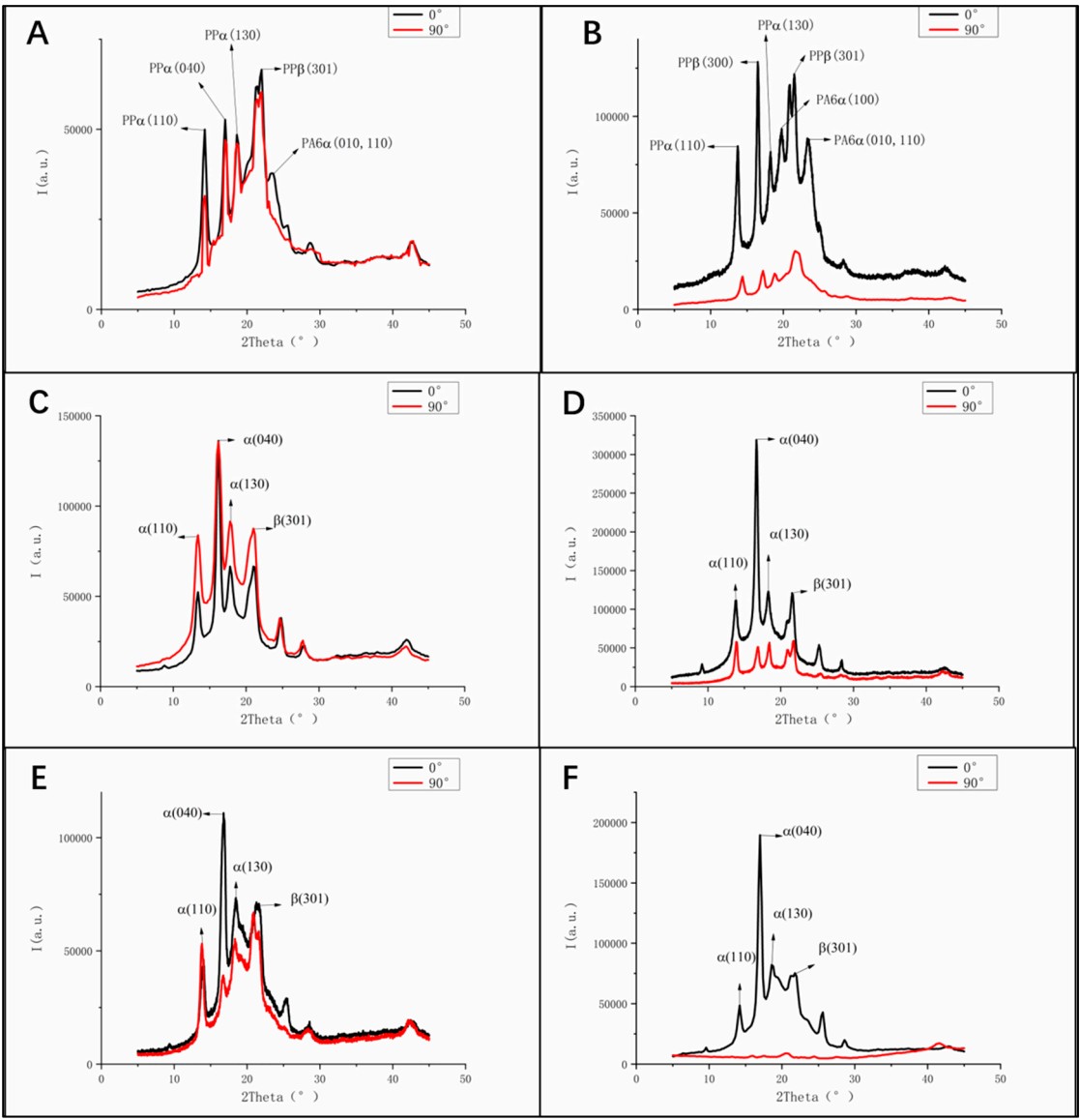

**Figure 3.** X-ray diffraction (XRD) spectra before and after PIF process, (**A**) PP/PA6, (**B**) PIF PP/PA6, (**C**) PP/POE, (**D**) PIF PP/POE, (**E**) PP/TPU, (**F**) PIF PP/ TPU.

**Table 1.** Table of crystal face statistics before and after PIF process.

| Samples | PP Crystalline |
|---------|----------------|
| PP/PA6 | α (110), (040), (130); β (301) |
| PIF PP/PA6 | α (110), (130); β (300), (301) |
| PP/POE | α (110), (040), (130); β (301) |
| PIF PP/POE | α (110), (040), (130); β (301) |
| PP/TPU | α (110), (040), (130); β (301) |
| PIF PP/TPU | α (110), (040), (130); β (301) |

It can be found that the α (040) crystal plane of PP changed to the β (300) crystal plane from PIF PP/PA6, and this phenomenon was not found in PP/POE and PP/TPU blends. This is due to PA6 have a higher glass transition temperature, resulting in no deformation of PA6 during PIF processing. The PP component is also subjected to the force of PA6 while being subjected to external pressure, resulting in the α crystal was transformed to the stronger β crystal, and it is consistent with the results obtained in the subsequent DSC spectrum. The general mechanism of material crystal deformation before and after PIF

treatment is shown in Figure 4. After PIF processing, different material crystals can achieve different degrees of deformation due to different strengths, and materials with different crystallinity which have different numbers of crystals per unit volume.

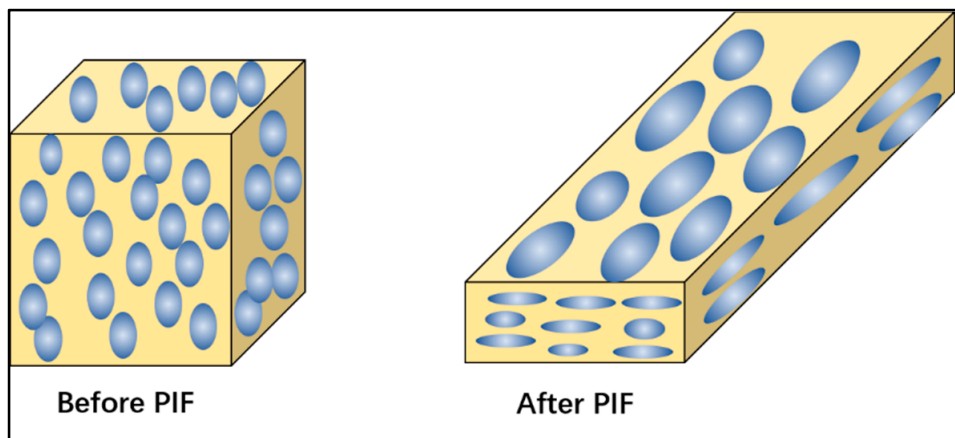

**Figure 4.** Crystal deformation mechanism diagram before and after PIF process.

### 3.2. The Effect of PIF Process on the Thermal Properties of PP Blends

As shown in Figure 5, DSC was used to study the melting behavior of the blends before and after PIF processing, and further study the effect of PIF processing on thermal properties. Tables 2–4 are the statistical results of the DSC curves (due to the crystallinity of POE and TPU is too low, the error of the two product peak areas of the same DSC curve is too large, so it is not counted).

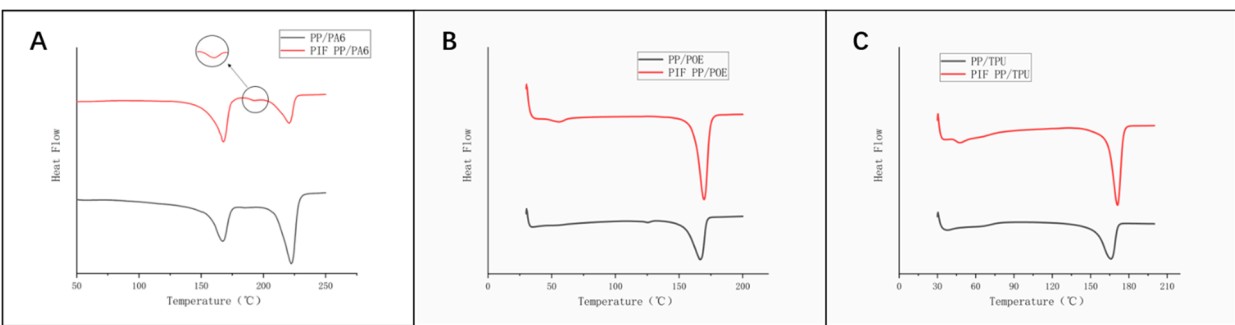

**Figure 5.** Differential scanning calorimetry (DSC) spectra before and after PIF process, (**A**) PP/PA6 and PIF PP/PA6, (**B**) PP/POE and PIF PP/POE, (**C**) PP/TPU and PIF PP/TPU.

**Table 2.** Thermal properties of PP/PA6 blends before and after PIF.

| Samples | $T_{m, PP}$ (°C) | $T_{m, PA6}$ (°C) | $T_{onset,PP}$ (°C) | $T_{onset,PA6}$ (°C) | $X_{C, PP}$ (%) | $X_{C, PA6}$ (%) |
|---|---|---|---|---|---|---|
| PP/PA6 | 165.52 | 219.76 | 154.56 | 208.51 | 16.56 | 30.32 |
| PIF PP/PA6 | 167.82 | 220.53 | 158.89 | 220.53 | 43.94 | 20.54 |

**Table 3.** Thermal properties of PP/POE blends before and after PIF.

| Samples | $T_{m, PP}$ (°C) | $T_{m, POE}$ (°C) | $T_{onset,PP}$ (°C) | $T_{onset,POE}$ (°C) | $X_{C, PP}$ (%) |
|---|---|---|---|---|---|
| PP/POE | 166.34 | 44.25 | 154.34 | 42.74 | 55.72 |
| PIF PP/POE | 171.78 | 44.97 | 162.70 | 43.16 | 58.33 |

**Table 4.** Thermal properties of PP/TPU blends before and after PIF.

| Samples | $T_{m, PP}$ (°C) | $T_{m, POE}$ (°C) | $T_{onset,PP}$(°C) | $T_{onset,POE}$ (°C) | $X_{C, PP}$ (%) |
|---|---|---|---|---|---|
| PP/TPU | 165.74 | 46.15 | 153.71 | 42.58 | 46.49 |
| PIF PP/TPU | 170.74 | 47.63 | 163.32 | 42.67 | 53.35 |

It can be seen from Table 2 that compared with non-PIF blends, PIF blends have higher crystallinity and higher initial melting temperature (Tonset) and peak melting temperature (Tm). The reason for the peak shift may be that the crystal becomes more perfect after PIF processing. It can also be seen from Figure 5 that the melting peak after PIF treatment becomes narrower, which proves that the crystal is more perfect. In Figure 5A, a small peak has been observed after the PP melting peak. However, Figure 5B,C were not found, which is corroborated by the speculation in the XRD analysis. The increase in crystallinity is due to the fact that the spherical crystal form is forced to deform under the PIF process, which makes the thickness of the crystal thicker, resulting in a higher melting enthalpy. This phenomenon has also been reported in other papers [23]. Therefore, the DSC results indicate that the blends material after PIF has formed an ordered layered stack with better thermodynamic properties.

### 3.3. The Effect of PIF Process on the Mechanical Properties of PP Blends

Through the above discussion, it is found that the PIF process can have a significant impact on the microstructure of the blends, resulting in a corresponding change in the mechanical properties of the blends under pressure. Herein, the flexural proper-ties and impact properties of the material were tested (the PIF process has little effect on the tensile properties of the material, so it hasn't been tested). The bending strength of PIF PP/PA6 shown in Figure 6A reached 66 MPa, which was nearly 55% higher than that of PP/PA6. The flexural strength of the PP/POE and PP/TPU blends shown in Figure 6B,C after the PIF process increased by nearly 80%. Specific data statistics are shown in Table 5.

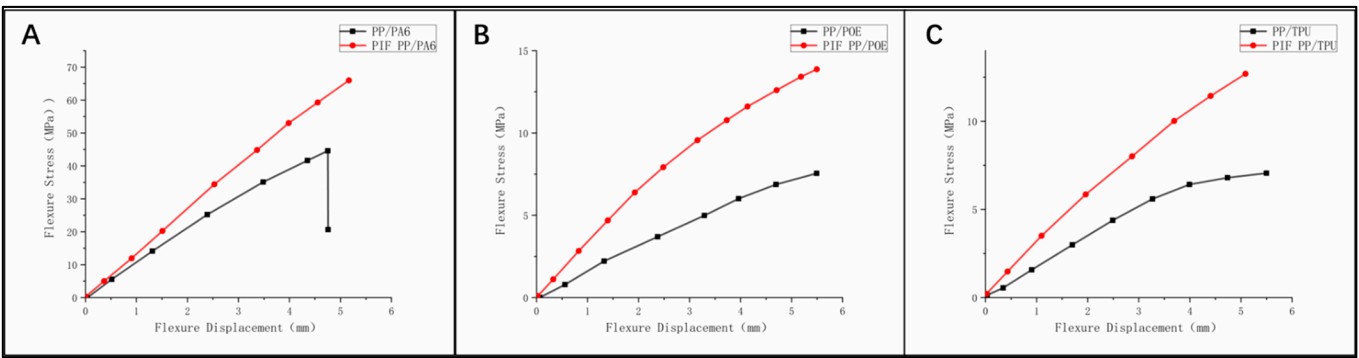

**Figure 6.** Bending properties of blends before and after PIF, (**A**) PP/PA6 and PIF PP/PA6, (**B**) PP/POE and PIF PP/POE, (**C**) PP/TPU and PIF PP /TPU.

**Table 5.** Bending strength and Increase ratio of blended materials.

| Sample | Bending Strength (MPa) | Increase Ratio (%) |
|---|---|---|
| PP/PA6 | 43.14 ± 0.5 | - |
| PIF PP/PA6 | 66.54 ± 0.3 | 54.24 |
| PP/POE | 7.67 ± 0.5 | - |
| PIF PP/POE | 13.84 ± 0.6 | 80.44 |
| PP/TPU | 7.06 ± 0.2 | - |
| PIF PP/TPU | 12.71 ± 0.5 | 80.03 |

The impact performance of the material is shown in Figure 7. After blending PA6 and PP, the toughness of the material decreased due to the increase in blends stiffness. Intuitively, after the PIF process, the impact strength of PP/PA6 is increased by nearly 400%. It can be seen from Figure 7 that blends have improved their impact strength after PIF treatment, the PP/TPU blends has increased by nearly 600% particularly. It shows that the PIF process has a significant toughening effect on the blends. The above results show that the micro-orientation and layered structure produced by the PIF process can effectively strengthen and toughen the blends at the same time. Specific data statistics are shown in Table 6.

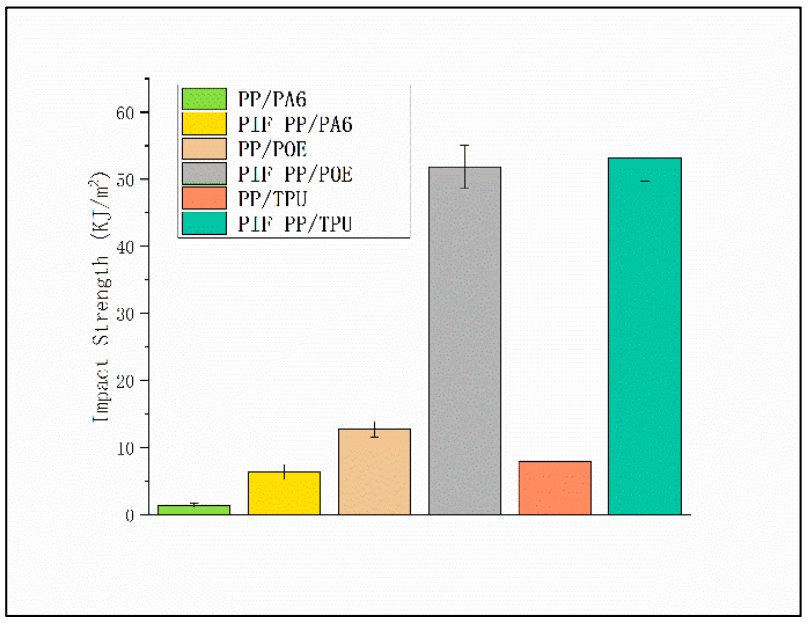

**Figure 7.** Impact performance of blends before and after PIF.

**Table 6.** Impact strength and Increase ratio of blended materials.

| Sample | Impact Strength (MPa) | Increase Ratio (%) |
|:---:|:---:|:---:|
| PP/PA6 | $1.36 \pm 0.3$ | - |
| PIF PP/PA6 | $6.33 \pm 0.7$ | 365 |
| PP/POE | $12.68 \pm 0.9$ | - |
| PIF PP/POE | $51.85 \pm 0.8$ | 308 |
| PP/TPU | $7.93 \pm 0.9$ | - |
| PIF PP/TPU | $53.21 \pm 0.7$ | 571 |

## 4. Conclusions

This paper studies the effect of PIF process on the microstructure, thermal and mechanical properties of PP blends. SEM, XRD, DSC characterization of PP/PA6, PIF PP/PA6, PP/POE, PIF PP/POE, PP/TPU, PIF PP/TPU blends show that the PIF process has a significant impact on the microstructure of PP/PA6 blends. At the same time, PIF processing significantly improves the mechanical properties of the blends. The bending strength of PP after PIF treatment is similar to PP/PA6, and the bending strength of PIF PP/PA6 reaches 66 MPa, which is 55% higher than PP/PA6. The bending strength of PP/POE and PP/TPU after PIF treatment has also been effectively improved. The impact strength of PIF PP/PA6 has increased by nearly 400%, which is almost the same as pure PP. The impact strength of PIF PP/POE and PIF PP/TPU increased by nearly 600%. The significant improvement in mechanical properties is due to the formation of a certain directional structure of the blends after the PIF process. The significant improvement in mechanical properties is due to the formation of a certain directional structure of the blends after the PIF process.

**Author Contributions:** Conceptualization, F.C. and P.L.; software, Y.J.; writing—review and editing, Y.F.; writing—original draft preparation, P.L.; raw material, J.Y.; discuss, S.R. All authors have read and agreed to the published version of the manuscript.

**Funding:** This research was funded by National Natural Science Foundation of China (No. 51803062), NSFC—Zhejiang Joint Fund for the Integration of Industrialization and Informatization (U1909219), Natural Science Foundation of Zhejiang Province (No. LY19E030007) and Opening Funding of State Key Laboratory of Structural Analysis for Industrial Equipment, Dalian University of Technology (No. GZ19115).

**Data Availability Statement:** No special permission is required to reuse all or part of article published by MDPI, including figures and tables. For articles published under an open access Creative Common CC BY license, any part of the article may be reused without permission provided that the original article is clearly cited. Reuse of an article does not imply endorsement by the authors or MDPI.

**Acknowledgments:** We thank the support of National Natural Science Foundation of China (No. 51803062), NSFC—Zhejiang Joint Fund for the Integration of Industrialization and Informatization (U1909219), Natural Science Foundation of Zhejiang Province (No. LY19E030007) and Opening Funding of State Key Laboratory of Structural Analysis for Industrial Equipment, Dalian University of Technology (No. GZ19115).

**Conflicts of Interest:** There is no conflict of interest.

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
