# Peer review of "The Influence of Pressure-Induced-Flow Processing on the Morphology, Thermal and Mechanical Properties of Polypropylene Blends"

_jcs, doi:10.3390/jcs5030064_

Round 1

Reviewer 1 Report

The effects of PIF process on different polymer blends have been studied in the paper. Relevant characterization tests are performed. Generally, paper is well written. However, it lacks in-depth discussion of the findings. For example, XRD results showed that the 0° plane (top surface as shown in Fig. 1) results before and after PIF process are almost similar. While 90° plane (side surface) XRD showed significant decrease in crystallinity. This should be discussed with proper scientific reasoning. Alternatively, it is concluded from DSC results that crystallinity increased after PIF process which is opposite to XRD results. I recommend accepting the paper but after substantial revision with more discussion.

The more specific comments are given here.

  1. Can you please clarify that what is the difference between pressure-induced formation (PIF) and hot press molding of polymers?
  2. In the Materials section, information of POE is missing.
  3. Lines 62 – 64, polymers were initially dried to eliminate the absorbed moisture. Please state the reasons of using different drying temperatures for each polymer. Similarly, clarify the reasons for using different processing temperature for polymer blends (lines 67- 69).
  4. Lines 71-72, are the dimensions of rectangular block and mold cavity correct? I wonder that how 9000 mm3 (50 x 30 x 6 mm) volume block will fill 54000 mm3 (150 x 30 x 12 mm) volume. I think mold thickness was 2 mm, not 12. Lastly, use rectangular block, instead of rectangular square.
  5. Lines 148 onwards, α (040) crystal of PP in PIF PP/PA6 sample changed to β (300) crystal. But discussion arguments are not coherent. First, author stated that higher glass transition temperature of PA6 resulted in no deformation of PP. But PP crystal phase already changed?
  6. The abbreviations PA6, POE, and TPU have been used. However, it is always recommended to write full names when referring for the first time in the manuscript.
  7. In Sec 3.3, authors stated that bending strength of PP has increased by 25% after the PIF process. This sample suddenly appears only in this section. There is no mention of solo PP sample in any previous section.

English corrections:

  1. Line 18, “At the same time, studies reveal that ….”. I assume that authors are talking about this manuscript, hence the phrase should be “study revealed that ….”.
  2. Line 35-36, “There are many studies have tried to ….”. Please revise.
  3. Lines 71-72, “Prepare a rectangular square ….”. Experimental setup should be explain in past tense such as rectangular square with …. was prepared. Same thing for lines 75-77.
  4. Line 157, delete “)” at the end of sentence.

Author Response

Thank you very much for your valuable comments. We have to postpone part of the work due to the arrival of the New Year, which has resulted in that we can only reply to you now. First, I wish you a happy new year, and then let me respond to your suggestions one by one.

Regarding the conflict between XRD and DSC data regarding crystallinity you mentioned, we will explain here: XRD pattern peaks are used to express the size and strength of crystal planes. Although the XRD signal of the 90° surface of the blend is weakened after PIF treatment. This can only prove that the crystal is squashed, which causes the 90° face of the crystal cross section to become narrow and long, the strength of the crystal face is reduced, and the crystal size is reduced. The more thorough the PIF, the greater the deformation of the crystal. This phenomenon is more obvious, but it has nothing to do with the crystallinity and only proves that the crystal is squashed. DSC is used to calculate crystallinity, so the two are complementary.

  1. The most significant difference between PIF and hot press molding is that the molding temperature of PIF is below the melting point and belongs to solid state pressing, while hot press molding generally belongs to melt pressing above the melting point.
  2. I am very sorry that we omitted the specific information of POE8400 due to our editorial negligence. Now it has been added to the article.
  3. For polymer drying and molding processing issues: Because different polymers have different water absorption properties, those with better water absorption require more time to dry, and those with poor water absorption require less time to dry. The decomposition temperature and melting point of various polymers are different, so drying also needs to be carried out at different temperatures. During processing, since PIF is a solid state, pressing must be carried out at the melting point of PP. However, if the temperature is too low, the material is brittle and difficult to be pressed. Therefore, we have determined the processing temperature in this article after many experiments. The processing temperature will change according to another component of the blend.
  4. We have corrected the description of the mold cavity2mm, and 12mm is the thickness of the entire mold rather than the thickness of the internal mold cavity.
  5. Questions about crystal morphology changes: We are sorry for this due to our editing error "PP will produce slippage during PIF due to its low melt strength and cannot change the crystalline form of PA6, so that the spherulites of PP cannot be changed." PA6 in is written as PP, and caused you trouble. It has now been corrected. The specific conclusion is that the PP melt cannot deform the PA6 spherulites due to the slip when the PP melt strength is too low and then it is pressed. Therefore, the PP bears more force and the PP crystal state changes.
  6. The full name of the material has been added in the abstract.
  7. Since the pure PP sample is only used to compare the mechanical strength of the blend, we did not conduct too much research on it. We also found its abruptness. The pure PP sample has been deleted and a statistical table has been added to the data of the mechanical strength map.
  8. The English words have been corrected according to your instructions: Line 18 "At the same time, research shows" is changed to "research shows". Change lines 35-35 "There are many studies trying" to "Many studies". Line 71-72 "Prepare a rectangular square" is changed to "Prepare a rectangle" and change the description to the past tense together with 75-77. The ")" at the end of line 175 has been deleted.

Finally, thank you for your valuable comments and suggestions for our work.

Reviewer 2 Report

The manuscript includes an interesting work focused on the analysis of influence of the pressure-induced-flow (PIF) processing technique on the morphology, thermal and mechanical properties of PP/PA6, PP/POE and PP/TPU. The objective of the study is clear. The methods and the analyses are correct. This manuscript is properly and logically organized. However, it is not clear in the manuscript why these blends are considered composites materials. In fact, in the article's bibliography and in the databases, they are considered as blends and they are only considered composites if talc, clay ... etc are incorporated. I think that it must be clarified in the title and it must be justified in the manuscript, especially for POE and TPU blends.

Please, add the complete name of the materials used. The acronyms POE and PTU must be explained at beginning and more details about the type of POE.

The introduction section should be expanded with a greater number of references about the main advantages have PIF processing. Please compare with other processing techniques.

In SEM images the PA6 is not appreciated, it would be interesting to show photos in which PA6 could be seen especially for the PP/PA6 samples. Figure 2 A is blurred or more magnification is necessary.

Figure 4 is confusing. In essence it explains that crystals are deformed, and a single generic scheme would be enough. However, it is confusing since it seems to indicate that after PIF a 2-layer material with a clear interface is produced. I think it should be improved and add clear explanations of the scheme.

The bending test are shown with figures, but the bending parameters are not shown. Please, add an additional Table with the bending parameters and the deviations.

How do the authors calculate the crystallinity of the blends PP/POE and PP/PTU?

Author Response

Thank you very much for your valuable comments. We have to postpone part of the work due to the arrival of the New Year, which has resulted in that we can only reply to you now. First, I wish you a happy new year, and then let me respond to your suggestions one by one.

  1. It is our negligence to refer to the blend as a composite material and we have corrected all of it as a blend material.
  2. The full name of the material has been added in the abstract. I am very sorry that we omitted the specific information of POE8400 due to our editorial negligence. Now it has been added to the article.
  3. Added a comparison between PIF and hot pressing in lines 46-51 of the introduction. References 31 and 32 describe the specific characteristics of PIF.
  4. Replace Figure 2A to make it clearer.
  5. Changed Figure 4 to use a more widely applicable mechanism diagram to explain crystal deformation, the material does form a layered structure after PIF, but this is a change in the amorphous region of the material. The crystalline part only undergoes crystal deformation, so for the mechanism of crystal deformation, we have eliminated the generation of layered structure, make the statement of crystal deformation simpler and more applicable.
  6. Statistical tables have been added to the mechanical properties section.
  7. The formula for material crystallinity has been given in line 106, see formula 1

Finally, thank you for your valuable comments and suggestions for our work.

Round 2

Reviewer 2 Report

The authors have considered all my suggestions.